# Eating Habits among US Firefighters and Association with Cardiometabolic Outcomes

**DOI:** 10.3390/nu14132762

**Published:** 2022-07-04

**Authors:** Andria Christodoulou, Costas A. Christophi, Mercedes Sotos-Prieto, Steven Moffatt, Stefanos N. Kales

**Affiliations:** 1Cyprus International Institute for Environmental and Public Health, Cyprus University of Technology, Limassol 3036, Cyprus; costas.christophi@cut.ac.cy; 2Department of Environmental Health, Harvard T.H. Chan School of Public Health, Boston, MA 02115, USA; mercedes.sotos@uam.es (M.S.-P.); stefokali@aol.com (S.N.K.); 3Department of Preventive Medicine and Public Health, School of Medicine, Universidad Autónoma de Madrid, 28029 Madrid, Spain; 4Instituto de Investigación Sanitaria Hospital Universitario La Paz (IdiPaz), 28029 Madrid, Spain; 5Biomedical Research Network Centre of Epidemiology and Public Health (CIBERESP), Carlos III Health Institute, 28029 Madrid, Spain; 6IMDEA-Food Institute, CEI UAM + CSIC, 28049 Madrid, Spain; 7National Institute for Public Safety Health, Indianapolis, IN 46204, USA; steven.moffatt@ascension.org; 8Department of Occupational Medicine, Cambridge Health Alliance, Harvard Medical School, Cambridge, MA 02145, USA

**Keywords:** Mediterranean diet, Mediterranean diet scores, dietary patterns, cardiometabolic risk

## Abstract

Cardiovascular disease is the leading cause of on-duty mortality among firefighters, with obesity as an important risk factor. However, little is known regarding the dietary patterns which are characteristic in this population and how these patterns relate to cardiometabolic outcomes. The aim of this study was to identify the dietary patterns of US firefighters and examine their association with cardiometabolic outcomes. The participants (*n* = 413) were from the Indianapolis Fire Department, and were recruited for a Federal Emergency Management Agency (FEMA)-sponsored Mediterranean diet intervention study. All of the participants underwent physical and medical examinations, routine laboratory tests, resting electrocardiograms, and maximal treadmill exercise testing. A comprehensive food frequency questionnaire was administered, and dietary patterns were derived using principal component analysis. The mean body mass index (BMI) was 30.0 ± 4.5 kg/m^2^ and the percentage of body fat was 28.1 ± 6.6%. Using principal component analysis, two dietary patterns were identified, namely a Mediterranean diet and a Standard American diet. Following the adjustment for gender, BMI, maximal oxygen consumption (VO_2_ max), max metabolic equivalents (METS), age, and body fat percent, the Mediterranean diet was positively associated with high-density lipoprotein (HDL) cholesterol (β = 1.20, *p* = 0.036) in linear regression models. The Standard American diet was associated with an increase in low-density lipoprotein (LDL) cholesterol (β = −3.76, *p* = 0.022). In conclusion, the Mediterranean diet was associated with more favorable cardiometabolic profiles, whereas the Standard American diet had an inverse association. These findings could help in providing adequate nutrition recommendations for US firefighters to improve their health.

## 1. Introduction

Firefighting is a hazardous occupation, and even though one might think that on-duty mortality among firefighters results from burns or smoke inhalation, the most frequent cause is sudden cardiac death (SCD) due to the underlying cardiovascular disease (CVD) [1]. On-duty fatalities in the US fire service account for almost half of all deaths and are due to SCD, strokes, aneurysms, and other CVD-related conditions. Furthermore, for every on-duty CVD-related death, there are an estimated 17 nonfatal on-duty CVD events [2,3,4]. Therefore, CVD is not only a leading cause of mortality among firefighters, but also a major cause of morbidity and resulting disability. Even though the cardiometabolic health of firefighters is better than the average US citizen, there is a decrease in the cardiometabolic health of male firefighters. Among female firefighters, cardiometabolic health shows a steady decrease, as well [5].

Several risk factors have been associated with the risk of CVD, including obesity, hypertension, and high cholesterol levels [6]. Obesity, which has negative effects on the fitness and performance of firefighters, is also shown to be associated with an increased risk of CVD, blood pressure, glucose metabolism, sleep apnoea, and cardiac enlargement [7].

Several population-based studies among volunteer and career firelighters have shown that the rise in obesity prevalence is not the result of an increase in muscle mass [8], but rather an increase in body fat [8]. This is an issue that affects younger firefighters, as well as middle-aged and older firefighters. Consequently, it is a problem that is not recognized to its full extent [9].

In this paper, we explore the reasons for the increase in obesity rates. A number of recent studies have shown that the difference between obese and non-obese firefighters is the increased consumption of sugary drinks and fast food [10,11]. These findings are consistent with other population-based studies, which suggest that switching the dietary patterns of people will have a large impact on their health. Shift work and uncontrollable mealtimes, which are the norm among firefighters, also tend to increase the consumption of sugary drinks and fast food, with a greater proportion of calories from fat [12,13].

One of the most well-accepted diets in the reduction of CVD risk is the Mediterranean diet. Mediterranean diets, traditionally followed by countries bordering the Mediterranean Sea, are rich in unrefined grains, fruits, vegetables, and olive oil, and contain a lower consumption of red meat and poultry [14]. Over the years, a large number of studies have demonstrated the effectiveness of the Mediterranean diet in the reduction of CVD mortality. The Mediterranean diet targets obesity, hypertension, diabetes, and metabolic syndrome, all of which are conditions associated with CVD [15,16,17]. Based on the clear benefits of the Mediterranean diet, it is recommended as one of the healthiest options in the US and other countries [18].

The first step in a nutritional intervention involves the identification of dietary patterns of the participant population. Dietary patterns are defined as “the quantity, variety or combination of different foods and beverage in a diet and the frequency with which they are habitually consumed” [19]. In a survey by Yang et al., obese firefighters were less likely to follow a dietary plan (25%) than normal-weight firefighters (33%). Among the 18 diets listed on the survey, 9% of the participants followed the Paleo diet, 8% a low-carbohydrate diet, and 4% a low-fat diet. Only 1% of the firefighters reported following the Mediterranean diet [20]. Similarly, in a study of 28 Swiss airport firefighters, the participants had an unbalanced diet with low-quality food choices and limited fiber intake [21].

Given that CVD is prevalent among firefighters, it is important to identify the dietary patterns of firefighters. Understanding the quality of different foods in the diet of firefighters can help us in providing scientific advice to improve food intake toward a healthier diet. The aim of this paper is to identify the dietary patterns of US firefighters and establish how these are associated with cardiometabolic outcomes in specific populations. Moreover, this would provide adequate recommendations to improve dietary interventions that target CVD and its related risk factors.

## 2. Materials and Methods

### 2.1. Study Participants

In this cross-sectional study, 413 firefighters were recruited from the Indianapolis Fire Department (IFD) (Indianapolis, IN, USA). The participants were enrolled as part of the study “Feeding America’s Bravest: Mediterranean Diet-Based Interventions to change firefighters’ Eating Habits and Improve Cardiovascular Risk Profiles” between November 2016 and April 2018. Recruitment and consent were carried out by the staff of the National Institute of Public Safety Health. Participants who did not complete baseline anthropometric measurements were excluded from the current analysis. More details on the study methodology and participant recruitment can be found in other literature [22].

### 2.2. Dietary Assessment

A validated 131-item food frequency questionnaire (FFQ) was administered to the participants [23]. The questionnaire collected information on the average frequency of consumption of each food item over the previous 12 months. Food items included dairy foods, fruits, vegetables, eggs, meat, breads, cereals, starches, beverages, sweets, baked goods, etc. [22].

### 2.3. Physical Activity

Physical activity was collected in participants’ assessments from the fire department medical examinations at Public Safety Medical (PSM) clinics, which was led by an IFD physician. The examinations included the collection of occupational, smoking, and medical history; a physical examination, including body mass index (BMI) and body fat measurements (using bioelectrical impedance); routine laboratory tests; resting electrocardiograms; and maximal treadmill exercise testing.

### 2.4. Outcome Assessment

At the initial visit, all of the participants underwent blood pressure and anthropometric assessments. An appropriately sized cuff was used to measure the resting blood pressure while the participants were in a seated position. BMI was recorded for all of the study subjects in kg/m^2^ and the percentage of body fat was estimated by a Bioelectrical Impedance Analyzer (BIA) [24,25].

The firefighters had their biochemical indices assessed at the medical examinations. We used the measurements collected from the date closest to the date of study consent and within the same 12-month period. Blood samples were collected after an overnight fast. Using ethylenediaminetetraacetic acid (EDTA) collection tubes, 15 mL of blood were collected. Plasma was frozen at −80 °C and the blood lipid profiles of the firefighters were determined using an automated high-throughput enzymatic analysis. Moreover, this analysis achieved the following values of the coefficient of variation: ≤3% for cholesterol and ≤5% for triglycerides, using the cholesterol assay kit and reagent (Ref: 7D62-21) and triglycerides assay kit and reagent (Ref: 7D74-21) by the ARCHITECT c System, Abbott Laboratories, Abbott Park, IL, USA.

### 2.5. Ethics Statement

The overarching “Feeding America’s Bravest” protocol was approved by the Harvard Institutional Review Board (IRB16-0170) ethics committee and is registered at Clinical Trials (NCT02941757). All of the participants provided signed informed consent for participation. The participants who met the criteria for enrollment were all informed about their right to decline participation or to withdraw at any time as per the Declaration of Helsinki, and the participants who decided to enroll gave full informed consent as per the protocol of the research [25].

### 2.6. Statistical Analysis

The principal component analysis (PCA) was used to identify the dietary patterns of the firefighters at baseline. A scree test was used to identify the number of factors present. Loading factors were calculated after a varimax rotation to obtain uncorrelated components, which can be more easily interpretable. To obtain a clearer pattern, a cut-off of ≥ |0.2| in factor loadings was applied. Continuous characteristics were presented as mean ± standard deviation (SD), whereas categorical variables were reported as frequency (percentage) by tertiles of the identfied dietary patterns (low-medium-high) and compared using the ANOVA test or the chi-square test of independence, respectively. Linear regression models were used to examine the effect of dietary patterns on cardiometabolic outcomes, after adjusting for age, gender, BMI, body fat percent, max metabolic equivalents (METS), and oxygen consumption (VO_2_) max. The resulting beta coefficients, together with the corresponding standard errors and *p*-values, were presented. As a sensitivity analysis, dietary patterns were used in the models, both as continuous variables as well as in tertiles. All of the statistical analyses were performed using SAS version 9.3 (SAS Institute Inc., Cary, NC, USA). The alpha level of significance was set at 0.05 and all of the tests were two-tailed.

## 3. Results

### 3.1. Baseline Characteristics

A sample of 413 firefighters had complete data for analysis in the current study. Firefighters’ baseline characteristics are shown in Table 1. The vast majority of the participants were males (94%), with a mean age of 47.2 ± 8.2 years. The average BMI in the study population was 30.0 ± 4.5 kg/m^2^. The average METS score was 11.6 ± 5.5 and the mean total cholesterol was 196.9 ± 38.3 (mg/dL). The participation rate of the study was 95%.

### 3.2. Dietary Patterns

Scree tests of the PCAs indicated two distinct factors—a Standard American diet (SAD) and a Mediterranean diet (MD). In the analysis, the total number of food items was 148, out of which 96 had a loading factor above the pre-set cut-off of ≥ |0.2|. The Mediterranean diet included 57 of these items, and consisted of vegetables (raw spinach (0.673), romaine lettuce (0.601), fruits (peaches, apple), wine, nuts (walnuts), and rice, as shown in Table 2.

Thirty-nine items scored loading factors above |0.2| in the Standard American diet, and those included red meat (hamburger, pork), pasta, and sweets (brownies), as shown in Table 2.

### 3.3. Categorization of Participants in Accordance with the Dietary Pattern

The cross-tabulation of the 413 participants in tertiles of MD and SAD are shown in Table 3. Several participants had both MD and SAD scores low as well as both MD and SAD high.

### 3.4. Association of Dietary Patterns with Cardiometabolic Outcomes

Participants in the highest tertile of Western diet were significantly worse in terms of weight, HDL cholesterol, cholesterol ratio, and triglycerides scores (Table 4). However, no significant differences were observed among the tertiles of Mediterranean diet.

The associations of dietary patterns with cardiometabolic outcomes are shown in Table 5. In unadjusted regression models, a unitary increase in SAD was significantly associated with increases in total cholesterol (β = 4.58, *p* = 0.014), LDL cholesterol (β = 3.88, *p* = 0.017), whereas it was associated with a decrease in HDL cholesterol (β = −0.59, *p* = 0.292). Moreover, there was a significant association between MD and HDL cholesterol levels (β = 1.14, *p* = 0.045). After adjusting for age, gender, VO_2_ max, max METS, BMI, and body fat percent, SAD was significantly associated with a higher body fat percent (β = 0.02, *p* = 0.922) and cholesterol ratio (β = 0.12, *p* = 0.026), whereas it was associated with a decrease in HDL cholesterol (β = −0.292, *p* = 0.578). Furthermore, we observed an increase in cholesterol (β = 4.49, *p* = 0.015) and triglycerides (β = 5.83, *p* = 0.090), although the results were not statistically significant. Finally, MD was significantly associated with an increase in HDL cholesterol (β = 1.20, *p* = 0.036) in the adjusted analysis, whereas it was associated with a decrease in cholesterol ratio (β = −0.05, *p* = 0.358).

When tertiles of dietary patterns were considered, the high SAD tertile compared with the low tertile was associated with a decrease in HDL cholesterol, whereas it was associated with an increase in cholesterol ratio, BMI, and body fat. Similarly, medium vs. low tertile of SAD was associated with an increase in total and LDL cholesterol, cholesterol ratio, and glucose. However, when adjusted for other score (MD diet score for SAD tertiles and SAD diet score for MD tertiles), no statistically significant associations were observed for the tertiles of MD with cardiometabolic outcomes, as shown in Appendix A
Table A1.

## 4. Discussion

Our study identified two major dietary patterns among Indianapolis US firefighters—a Standard American diet and a Mediterranean diet. This was not surprising, given that the Western diet is one of the most common diets among US citizens and the Mediterranean diet is one of the most common diets among people pursuing a healthier lifestyle [20]. Several studies of focus groups found that firefighters have an unhealthy diet when eating at the firehouse, with large portions of food, unhealthy comfort foods, and second servings, whereas at the comfort of their home, they follow a healthier diet [24]. Moreover, several studies have previously argued that the majority of firefighters do not follow any specific dietary plan, although they may have their own routine in place in terms of eating habits. In our population, both dietary patterns share common food items, mainly fruits and vegetables. However, the MD is richer in vegetables, such as spinach, pepper, peas, and dairy, whereas the SAD is richer in meat and processed foods, such as beef, hamburger, bacon, sausage, etc. Sharing common food items and mainly fruit and vegetables could be attributed to the fact that people nowadays tend to eat a variety of different foods. Therefore, there is no surprise that people who consume a more Western diet tend to also eat fruits and vegetables as reflected in the dietary pattern.

Our analysis shows that the Mediterranean diet was associated with higher HDL cholesterol levels. These results are in agreement with those of other studies, suggesting that the Mediterranean pattern, which is characterized by high consumption of vegetables, is a “healthy” diet and can help lower cardiovascular risk [25]. Furthermore, greater consumption of fruits and vegetables, which is another element of the Mediterranean diet, was linked with higher fiber, folate, and potassium intakes. Dietary fiber may play a protective role against non-communicable diseases. Even though the mechanism for this is not fully understood, higher intakes of fruits and vegetables are strongly associated with lower CVD development [26,27].

In contrast, the Standard American diet was characterized by high consumption of red meat and sugary foods [28]. Meat and meat products were common constituents of the Standard American diet. In accordance with the US Department of Agriculture (USDA), consumers ate on average 100.8 kg of red meat and poultry in 2018 [29]. Meta-analysis studies showed that the greater consumption of processed meat was associated with 42% higher risk of developing coronary heart disease and 19% higher risk of diabetes. In our study, the Standard American diet was associated with an increase in LDL cholesterol and total cholesterol levels [30,31]. These results support the fact that high saturated fat diets were associated with worse cardiometabolic outcomes.

One limitation of this study is its cross-sectional nature, which does not allow us to infer causation. A second limitation concerns the items used in the food frequency questionnaire, which could be considered as common foods of the MD or SAD. In addition, they were included in the PCA analysis and resulted in a relatively low loading factor. Therefore, the pre-set cut-off was |0.2|. A third limitation concerns the very low number of female participants (6%). However, this reflects the current demographic of the US career fire service. One of the strengths of the study is that with the help and support of the IFD, the Indianapolis Local 416 fire station, and the recruited participants, we were able to collect all the necessary medical data required for the analysis. Another strength is that all our data were collected from the medical files of the participants with the help of a team of trained physicians, thus ensuring their validity.

## 5. Conclusions

In conclusion, this is one of the first studies in the US to report on specific dietary patterns among firefighters. These patterns were identified as the Mediterranean diet and the Standard American diet. In addition, the present findings confirmed and further strengthened the current knowledge regarding the positive associations of the Mediterranean diet and the negative associations of the Standard American diet on cardiometabolic outcomes. Further studies should investigate the role of diet in specific populations, as identifying the different diet components can assist in the creation of programs that improve the health of firefighters to save lives.

## Figures and Tables

**Table 1 nutrients-14-02762-t001:** Baseline characteristics.

Characteristics	Overall
Males	390 (94%)
Age (years)	47.2 (8.2)
Smoking	10 (3.9%)
Alcohol (units per week)	12.81 (20.18)
Height (m)	1.79 (0.07)
Weight (kg)	96.8 (17.4)
BMI (kg/m^2^)	30 (4.5)
% body fat (%)	28.1 (6.6)
Max METS	11.6 (5.5)
Est. VO_2_ max	42.1 (4.97)
Diastolic BP (mmHg)	78.3 (6.1)
Systolic BP (mmHg)	123.4 (8.8)
Cholesterol (mg/dL)	196.9 (38.3)
HDL cholesterol (mg/dL)	49.2 (11.4)
LDL cholesterol (mg/dL)	122.7 (33.1)
Cholesterol Ratio	4.20 (1.30)
Triglycerides (mg/dL)	124.5 (75.65)
Glucose (mg/dL)	100.0 (20.5)

BMI, body mass index; METS, metabolic equivalents; Est. VO_2_, estimated oxygen consumption; BP, blood pressure; HDL, high-density lipoprotein; LDL, low-density lipoprotein.

**Table 2 nutrients-14-02762-t002:** Dietary Patterns.

Dietary Patterns-Mediterranean Diet	
Food Item	Loading Factor
Raw spinach	0.673
Romaine lettuce	0.601
Beans	0.599
Cantaloupe	0.564
Peaches	0.554
Cooked spinach	0.553
Celery	0.543
Peppers	0.538
Raw carrot	0.476
Cooked carrot	0.475
Orange winter squash	0.465
Orange	0.454
Blueberries	0.453
Peas	0.433
Apricot	0.432
Low Carb Bars	0.424
Low Calorie Beverage without Caffeine	0.422
Cream Cheese	0.404
Avocado	0.402
Salsa	0.399
Sweet potato	0.397
Tomato	0.396
Energy Bars	0.391
Banana	0.390
Rye bread	0.390
Cabbage	0.381
Kale	0.369
Apple	0.368
String Beans	0.334
Olive oil	0.328
Cottage Ricotta cheese	0.315
Tomato sauce	0.300
Tofu	0.290
English muffin/Bagels/Rolls	0.287
Raisin grapes	0.286
Bacon	0.266
White wine	0.265
Breakfast bars	0.260
Other nuts (other than peanuts/walnuts)	0.259
Red wine	0.258
Potato	0.256
Zucchini	0.252
Margarine	0.251
Brown rice	0.246
Pure Butter	0.236
Eggs	0.228
Pretzel	0.215
Yogurt	0.212
Tomato juice	0.212
Apple juice	0.211
Plain yogurt	0.210
Peanut butter	0.209
Coffee	0.209
Cooked cereal (other than oatmeal)	0.206
Popcorn	0.204
Walnuts	0.201
Fresh Fried Potatoes	0.200
**Dietary patterns-Standard American Diet**	
**Food Item**	**Loading Factor**
Other Fish (other than dark meat fish)	0.900
Corn	0.877
Dark Meat Fish	0.877
Brussels	0.863
Chichen with skin	0.835
Cauliflower	0.831
Broccoli	0.811
Mayonnaise	0.754
Sweets	0.752
Bologna	0.728
Ice lettuce	0.701
Cooked Oatmeal	0.699
Orange juice	0.694
Strawberries	0.673
Mixed vegetables	0.647
Whole grain bread	0.582
Chicken sandwich	0.566
Beef Burger sandwich	0.560
Cooked Onions	0.476
Hotdog	0.475
Chicken without skin	0.458
Breaded Fish Pieces	0.452
Oil and vinegar	0.443
Onions as garnish	0.433
White rice	0.370
Beef	0.304
Brownie	0.301
Chicken Hot Dog	0.301
Processed meat	0.296
Hamburger	0.294
Cooked Shrimp	0.292
Pork	0.291
Carbonated drink with Sugar but no Caffeine	0.288
Ready-made Pie	0.288
White bread	0.281
Punch	0.273
Oat Bran	0.271
Other fresh juice (other than orange and grapefruit)	0.241
Pasta	0.235
Cake	0.200

**Table 3 nutrients-14-02762-t003:** Different dietary patterns of participants.

Standard American Diet	Mediterranean Diet			Total
	Low	Medium	High	
**Low**	76	45	15	136
**Medium**	44	54	39	137
**High**	17	37	86	140
**Total**	137	136	140	413

**Table 4 nutrients-14-02762-t004:** Cardiometabolic characteristics.

Characteristic	Standard American Diet				Mediterranean Diet			
	Low	Medium	High	*p*-Value	Low	Medium	High	*p*-Value
Height (m)	1.79 (0.07)	1.79 (0.07)	1.80 (0.07)	0.155	1.79 ± 0.07	1.79 ± 0.07	1.80 ± 0.07	0.779
Weight (kg)	93.3 (16.20)	96.02 (15.71)	101.11 (19.18)	<0.001	96.00 ± 14.80	96.77 ± 19.00	97.80 ± 18.33	0.698
body fat (%)	27.24 ± 6.95	27.89 ± 6.17	29.03 ± 6.70	0.084	28.012 ± 5.71	28.31 ± 6.87	27.74 ± 7.30	0.778
Max METS	12.16 ± 9.24	11.29 ± 1.31	11.24 ± 2.43	0.318	11.96 ± 9.12	11.28 ± 1.58	11.45 ± 2.44	0.578
VO_2_ max	42.58 ± 5.31	42.19 ± 4.69	41.58 ± 4.89	0.258	42.19 ± 4.70	42.01 ± 5.06	42.13 ± 5.18	0.960
Diastolic BP (mmHg)	78.21 ± 5.86	78.01 ± 6.08	78.80 ± 6.19	0.539	78.62 ± 5.67	78.70 ± 6.08	77.72 ± 6.38	0.338
Systolic BP (mmHg)	123.05 ± 8.86	123.17 ± 8.72	123.98 ± 8.93	0.639	123.32 ± 8.38	123.41 ± 9.14	123.49 ± 9.01	0.987
Cholesterol (mg/dL)	193.15 ± 38.28	199.59 ± 39.99	197.55 ± 37.22	0.363	196.84 ± 39.51	194.77 ± 36.96	198.69 ± 39.06	0.700
HDL cholesterol (mg/dL)	51.52 ± 12.00	48.84 ± 11.19	47.29 ± 10.66	0.008	48.54 ± 10.64	49.10 ± 12.14	49.90 ± 11.37	0.612
LDL cholesterol (mg/dL)	119.03 ± 31.70	126.90 ± 33.73	122.15± 33.43	0.150	122.83± 31.67	122.01 ± 33.96	123.32 ± 33.67	0.947
Cholesterol Ratio	3.95 ± 1.52	4.24 ± 1.08	4.37 ± 1.25	0.023	4.27 ± 1.63	4.16 ± 1.16	4.14 ± 1.06	0.647
Triglycerides (mg/dL)	106.79 ± 59.75	120.64 ± 64.14	145.20 ± 93.08	<0.001	124.80 ± 87.07	119.56 ± 64.83	129.11 ± 76.01	0.581
Glucose (mg/dL)	97.82 ± 18.02	102.44 ± 27.60	99.77± 13.20	0.180	101.20 ± 20.46	98.19 ± 19.10	100.64 ± 21.78	0.440

**Table 5 nutrients-14-02762-t005:** Association of dietary patterns with cardiometabolic outcomes.

	Unadjusted Models						Adjusted Models *					
Outcome	Standard American Diet			Mediterranean Diet			Standard American Diet			Mediterranean Diet		
	β	se	*p*	β	se	*p*	β	se	*p*	β	se	*p*
BMI	0.23	0.23	0.292	0.15	0.24	0.527	0.02	0.19	0.922	0.30 **	0.21	0.150
Body Fat	0.45	0.33	0.166	−0.22	0.35	0.537	0.26	0.27	0.331	0.02 **	0.20	0.943
Cholesterol	4.58	1.86	0.014	0.85	1.92	0.657	4.49	1.84	0.015	1.18	2.02	0.559
HDL cholesterol	−0.59	0.56	0.29	1.14	0.57	0.045	−0.292	0.52	0.578	1.20	0.57	0.036
LDL cholesterol	3.88	1.61	0.017	−0.03	1.66	0.985	3.76	1.63	0.022	−0.31	1.79	0.865
Cholesterol ratio	0.14	0.06	0.033	−0.08	0.07	0.244	0.12	0.05	0.026	−0.05	0.06	0.358
Triglycerides	7.73	3.69	0.037	−0.09	3.79	0.982	5.83	3.43	0.090	1.37	3.75	0.715
Glucose	−0.53	1.00	0.594	−1.05	1.02	0.305	−0.97	0.94	0.506	−0.01	1.03	0.990

* Adjusted for gender, max METS, VO_2_ max, age, BMI, and body fat percent. ** Adjusted for gender, max METS, VO_2_ max, age. Se, standard error.

## Data Availability

The authors will make deidentified raw data set available upon reasonable requests.

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
