# Peer review of "Eating Habits among US Firefighters and Association with Cardiometabolic Outcomes"

_nutrients, 2022, doi:10.3390/nu14132762_

Round 1

Reviewer 1 Report

Authors used results of physical examinations and dietary survey among US firefighters and examined associations between dietary pattern (Mediterranean diet and standard American diet) and cardiovascular disease risk factors (BP, serum lipids, and blood glucose).

CVD risks have been reported be higher in firefighters and there may be significance, however, they did not included the classis risk factors known to be linked to those CVD risk factors; i.e., smoking habit (is linked to HDL cholesterol), alcohol consumption (is linked to HDL cholesterol and triglyceride). The statistical analyses lacking in those lifestyle factors are not sufficient to support the authors’ conclusion.

  1. Materials and Methods

Study participants

 “This cross-sectional study included firefighters from the Indianapolis Fire Department (IFD).”

Authors should present the participation rate.

Statistical analysis

“were reported as frequency (percentage) by tertiles of the identified dietary patterns (Low - Medium - High)”

It seemed that authors divided the participants twice (in two ways), how they fit in Mediterranean diet, and how they fit in standard American diet, using dietary pattern scores? Authors should describe these process more precisely and specifically.

Maybe, those who are high in Mediterranean diet were low in standard American diet? Authors can present a cross tabulation.

Table 2. 1 and 2

Many description of “food item” make no sense; i.e., cr.ch, a.j, p.bu, ff.pot. Maybe they are the variable names in the statistical dataset?

My impression is that Mediterranean diet is high in intakes of fruits, vegetables, fish, olive oil, pasta, and standard American diet is high in fat and fatty meat products. In the Table 2.1, bacon, margarine, butter are listed as high in loading factor with Mediterranean diet, and in the Table 2.2, “oth.fish” (other fish?), shrimp, pasta are listed as high in loading factor with standard American diet. They seems somewhat weird.

Author Response

Thank you for your constructive feedback.

Regarding your comments:

Comment 1.The statistical analyses lacking in those lifestyle factors (smoking and alcohol consumption) are not sufficient to support the authors’ conclusion.

Response:

There were only n=11 smokers in our group of participants and adding this in the analysis had no significant impact to the results. Alcohol consumption was also very low.

Comment 2.

Study participants

 “This cross-sectional study included firefighters from the Indianapolis Fire Department (IFD).”

Authors should present the participation rate.

Response:

Participation rate was 95%.

Comment 3.Authors can present a cross tabulation.

Response:

Please see cross-tabulation table

SAD

MD

Total

Low

Medium

High

Low

81

46

15

142

Medium

44

58

40

142

High

17

38

87

142

Total

142

142

142

426

Comment 4:Many description of “food item” make no sense; i.e., cr.ch, a.j, p.bu, ff.pot. Maybe they are the variable names in the statistical dataset?

Response: 

Thank you for your comment. We have now corrected all abbreviated food items.

Comment 5: My impression is that Mediterranean diet is high in intakes of fruits, vegetables, fish, olive oil, pasta, and standard American diet is high in fat and fatty meat products. In the Table 2.1, bacon, margarine, butter are listed as high in loading factor with Mediterranean diet, and in the Table 2.2, “oth.fish” (other fish?), shrimp, pasta are listed as high in loading factor with standard American diet. They seems somewhat weird.

Response:

We have also noticed that. We have now noted this in the discussion and added the following text:

Sharing common food items, such as fruits and vegetables, could be attributed to the fact that people nowadays tend to eat more of everything. Therefore, it is not surprising that people who eat a more westernized diet, high in calories, eat more fruits and vegetables as well, and this is reflected in the diet pattern.

Reviewer 2 Report

Congratulations on the work done. I have a few questions regarding your manuscript. 

  1. Do you have any information about comorbidities in subject before including them in the study?
  2. Should not you exclude women from your analysis?

It seems interesting to also analyze the frequency and regularity of meals, it might improve the value of your work. 

Author Response

Thank you for your constructive feedback.

Regarding your comments:

comment 1: Do you have any information about comorbidities in subject before including them in the study?

Response: Thank you for your suggestion. Unfortunately, we do not have information about comorbidities in our participants before including them in the study.

comment 2: Should not you exclude women from your analysis?

Response:

Thank you for your comment. When only men were included in the analysis results did not change in any significant ways so we decided to keep women in the analysis and have a larger sample which also represents the firefighters’ population more closely. We have also added the following text in the Discussion section:

Another limitation was the low number of female participants (6%). However, this reflects the current demographic of the US career fire service.

Comment 3: It seems interesting to also analyze the frequency and regularity of meals, it might improve the value of your work. 

Rsponse:Thank you for the nice suggestion. The issue of frequency and regularity of meals as well as timing of meals, given the shift work performed by firefighters, would be of great interest but unfortunately this information was not part of the current study. We are working on other studies which are trying to look at this issue.

Round 2

Reviewer 1 Report

“There were only n=11 smokers in our group of participants and adding this in the analysis had no significant impact to the results. Alcohol consumption was also very low.”

 Smoking habit and drinking habit are both basic and classic risk factors for CVD. Authors should include them in the Table irrespective of the results of analysis. It is like showing participant’s age or sex. Please include frequency of smokers and drinkers (or average consumption of alcohol).

 “Participation rate was 95%.”

 I could not find the participant rate in the manuscript. The participant rate is important in epidemiological study to evaluate representativeness of the study. Usually the rate is presented in the head RESULTS section.

 “Please see cross-tabulation table”

As I had imagined, a lot of participants were categorized as “high in MD and high in SAD”. What does this mean? What happens if authors analyzed data comparing MD tertile groups adjusting SAD score, or comparing SAD tertile groups adjusting MD score? Please describe the interpretation of the “overlapping” of the two dietary patterns”.

This cross table is important in interpreting the results. Please include this table in the manuscript.

 ”We have now corrected all abbreviated food items.”

 Thank you for the edits. However, I still do not understand some of the food descriptions.

For examples: “Local”; “zone.bar”; “cr.ch”; “otherfish” should be “other fish”; what does “Different kind of fish” mean?; “Orangejuicecalc”; “Bologna”; “chickensadwich” (chicken sandwich?); “No chicken”; “onions” and “Onions” (two times in the table”); “chicken.dog”; “shrimp.cooked”; what does “Other carbohydrates” measn?; “pie.comm”; “oat.bran”.

Please be sure to start the food description with capital letters. This is a scientific article and authors should be careful in use of words.

 The title of the manuscript is “Mediterranean Diet as an Eating Pattern among US Firefighters and its association with cardio-metabolic outcomes”, but authors classified participants two times, with MD scores and SAD scores which is contrary to the title. And some of the participants were high in both MD and SAD (tangled situation). It might be wise for authors to change the title and use the MD scores and SAD scores as adjusting factors to detect independent associations with CVD risk factors.

Author Response

Thank you for your comments.

Comment 1: “There were only n=11 smokers in our group of participants and adding this in the analysis had no significant impact to the results. Alcohol consumption was also very low.” Smoking habit and drinking habit are both basic and classic risk factors for CVD. Authors should include them in the Table irrespective of the results of analysis. It is like showing participant’s age or sex. Please include frequency of smokers and drinkers (or average consumption of alcohol).

Response 1: The frequency of smoker and drinkers have now been included in the results.

Comment 2: “Participation rate was 95%.” I could not find the participant rate in the manuscript. The participant rate is important in epidemiological study to evaluate representativeness of the study. Usually the rate is presented in the head RESULTS section.

Response 2: Participation rate has now been included in the results section. Comment 3: As I had imagined, a lot of participants were categorized as “high in MD and high in SAD”. What does this mean? What happens if authors analyzed data comparing MD tertile groups adjusting SAD score, or comparing SAD tertile groups adjusting MD score? Please describe the interpretation of the “overlapping” of the two dietary patterns”.

Response 3: In Table 1A in the Appendix section the tertiles were also adjusted with the diet score and the results are presented.

Comment 4:This cross table is important in interpreting the results. Please include this table in the manuscript.

Response 4: a new section was added to the results with the cross-tabulation table (section 3.3 Table 3)

Comment 5: Thank you for the edits. However, I still do not understand some of the food descriptions. For examples: “Local”; “zone.bar”; “cr.ch”; “otherfish” should be “other fish”; what does “Different kind of fish” mean?; “Orangejuicecalc”; “Bologna”; “chickensadwich” (chicken sandwich?); “No chicken”; “onions” and “Onions” (two times in the table”); “chicken.dog”; “shrimp.cooked”; what does “Other carbohydrates” measn?; “pie.comm”; “oat.bran”. Please be sure to start the food description with capital letters. This is a scientific article and authors should be careful in use of words.

Response 5: Further corrections were made.

Comment 6:The title of the manuscript is “Mediterranean Diet as an Eating Pattern among US Firefighters and its association with cardio-metabolic outcomes”, but authors classified participants two times, with MD scores and SAD scores which is contrary to the title. And some of the participants were high in both MD and SAD (tangled situation). It might be wise for authors to change the title and use the MD scores and SAD scores as adjusting factors to detect independent associations with CVD risk factors.

Response 6: The title of the manuscript has now been changed to 'Eating Patterns Among US Firefighters and Association with Cardio-metabolic Outcomes’.